# A role for subducting clays in the water transportation into the Earth's lower mantle

Yoonah Bang [1,2], Huijeong Hwang[3,4], Hanns-Peter Liermann [3], Duck Young Kim [5,6], Yu He[5,7], Tae-Yeol Jeon[8], Tae Joo Shin [9], Dongzhou Zhang [10,11], Dmitry Popov[12] & Yongjae Lee [1] ✉

Subducting sedimentary layer typically contains water and hydrated clay minerals. The stability of clay minerals under such hydrous subduction environment would therefore constraint the lithology and physical properties of the subducting slab interface. Here we show that pyrophyllite ($Al_2Si_4O_{10}(OH)_2$), one of the representative clay minerals in the alumina-silica-water ($Al_2O_3$-$SiO_2$-$H_2O$, ASH) system, breakdowns to contain further hydrated minerals, gibbsite ($Al(OH)_3$) and diaspore ($AlO(OH)$), when subducts along a water-saturated cold subduction geotherm. Such a hydration breakdown occurs at a depth of ~135 km to uptake water by ~1.8 wt%. Subsequently, dehydration breakdown occurs at ~185 km depth to release back the same amount of water, after which the net crystalline water content is preserved down to ~660 km depth, delivering a net amount of ~5.0 wt% $H_2O$ in a phase assemblage containing δ-$AlOOH$ and phase Egg ($AlSiO_3(OH)$). Our results thus demonstrate the importance of subducting clays to account the delivery of ~22% of water down to the lower mantle.

Subduction zones impose an important constraint on the global circulation of $H_2O$ between the surface and the interior of the Earth[1,2]. A subducting slab is generally composed of sediments, oceanic crust, and a part of the upper mantle in about 1:6:50 volumetric ratio[3], featuring distinctive petrological and geochemical processes in each layer. While the net $H_2O$ flux into the Earth's deep interior is dominated by the oceanic crust and upper mantle composed of hydrated altered lithology such as serpentinites[2], water transportation and/or mineral-water interaction by the topmost sedimentary layer is of significant importance as it occurs along the interface between the subducting slab and the overriding mantle wedge with a high water-to-sediment ratio, e.g., by about 2:3 ($0.9 \times 10^{15}$ g/yr:$1.4 \times 10^{15}$ g/yr) in the fluid-rich sediment in the vicinity of subduction trench[4]. There are various fluid migration pathways in a subduction zone, i.e., across the slab-mantle interface into the mantle wedge as well as within the subducting slab itself (see below)[5]. Many studies have revealed the existence of a narrow and highly strained mixing zone, so-called, subduction channel, between the subducting slab and the overlying mantle wedge[6,7]. The slab-mantle interface preserves metastable hydrous phases and/or fluids generated by a sequence of dehydration reactions during subduction[6]. Such a slab-mantle interface extends from the Earth's surface to depths exceeding ~100 km, i.e., beneath volcanic arcs

[1]Department of Earth System Sciences, Yonsei University, Seoul 03722, Republic of Korea. [2]Korea Atomic Energy Research Institute (KAERI), Daejeon 34057, Republic of Korea. [3]Photon Sciences, Deutsches Elektronen-Synchrotron (DESY), Hamburg 22607, Germany. [4]School of Earth Sciences and Environmental Engineering, Gwangju Institute of Science and Technology, Gwangju 61005, Republic of Korea. [5]Center for High Pressure Science & Technology Advanced Research, Shanghai 201203, China. [6]Division of Advanced Nuclear Engineering, Pohang University of Science and Technology, Pohang 37673, Republic of Korea. [7]Key Laboratory of High-Temperature and High-Pressure Study of the Earth's Interior, Institute of Geochemistry, Chinese Academy of Sciences, Guiyang, Guizhou 550081, China. [8]Pohang Accelerator Laboratory, POSTECH, Pohang 37673, Republic of Korea. [9]Graduate School of Semiconductor Materials and Devices Engineering, Ulsan National Institute of Science and Technology (UNIST), Ulsan 44919, Republic of Korea. [10]Hawaii Institute of Geophysics and Planetology, University of Hawaii at Manoa, Honolulu, HI 96822, USA. [11]GSECARS, University of Chicago, Chicago, IL 60439, USA. [12]High Pressure Collaborative Access Team, X-ray Science Division, Argonne National Laboratory, Lemont, IL 60439, USA. ✉e-mail: yongjaelee@yonsei.ac.kr

toward the deeper mantle[6,7]. Within the subducting slab, fluid flux becomes channelized into veins as prograde mineral reactions form open fractures due to negative volume change and associated brittle deformation[8]. Especially, water could be trapped in a subducting slab as free fluid and subsequently be liberated through some mechanical processes, phase transformations, or differences in dihedral angle due to changing P-T conditions or fluid composition[9]. In the slab-mantle interface, fluid flux is constrained by the degree of viscous deformation and becomes channelized in ductile shear zones to be aligned to the slab-mantle interface[5]. In the mantle wedge, fluid flux occurs through pervasive network of microfractures and grain boundaries owing to retrograde and rehydration reactions accompanying positive volume change[10]. According to Schmidt and Poli[11], a substantial portion, ranging from 18 to 37%, of the total subducted water will be lost by the subarc depths of ~80–150 km through dehydration reactions. The released fluid could be trapped as an interstitial component and subsequently transported by the descending slab to depths of at least ~150 km[12]. Therefore, fluids in subduction zones could persistently infiltrate into the overlying layers and mantle wedge[8]. Accordingly, within a subducting oceanic plate, metasediments will interact with fluids originating from the dehydration processes in both the crust and mantle layers[8].

In a subducting slab, the sediments themselves are mainly composed of hydrated clay minerals abundant in alumina and silica of ~17–21 wt% and ~55–63 wt%, respectively[13–16]. In contrast, a typical mid-ocean ridge basalt contains less alumina and silica to ~13–17 wt% and ~45–50 wt%, respectively, while these contents are further reduced to ~3–4 wt% and ~40–45 wt%, respectively, in the altered peridotite in the upper mantle[3,13,17]. Therefore the hydrated oceanic sediments can be represented by the simplified ternary system of $Al_2O_3$-$SiO_2$-$H_2O$ (ASH)[18,19] and thus suggested as the potential source materials for deep water transportation since, in recent studies, breakdown products in the ASH system have been demonstrated to have extended stabilities down to the lower mantle and the core-mantle boundary conditions (>150 GPa)[20–23]. However, it is surprising that high-pressure and high-temperature (HP-HT) studies of clay minerals are scarce, compared to those of crustal minerals and serpentines, although understanding the stabilities of clay minerals, especially along the water-rich subduction interface, would provide new insights into the origins of the ASH system and related deep $H_2O$ transport into the Earth[24,25]. In our previous work, we have demonstrated that subducting kaolinite $(Al_2Si_2O_5(OH)_4)$, one of the representative oceanic clay sediments in the ASH system, increases its $H_2O$ transport capacity via super-hydration, i.e., a counter-intuitive mineral transformation in which a hydrated mineral uptakes more water to form a further hydrated mineral; kaolinite becomes super-hydrated $(Al_2Si_2O_5(OH)_4 \cdot 3H_2O)$ at a depth of about 75 km along a water-rich cold subduction interface, which subsequently breaks down near 200 km depth to form other minerals in the ASH system[26]. This work called for the need for ree-valuating the overall impacts of subducting clays for the origins of the ASH minerals and water transportation into the deep Earth.

Pyrophyllite $(Al_2Si_4O_{10}(OH)_2)$ is a hydrous clay mineral consisting of a sheet of aluminum dioctahedra sandwiched between two layers of silicon tetrahedra, representing the so-called 2:1 clay mineral group in the ASH system. Similar to kaolinite, a representative clay for the 1:1 group[26], pyrophyllite does not possess interlayer cations nor water molecules at ambient conditions and hence is nominally 'non-expandable', making it an ideal candidate to examine the possible intercalations of water, i.e., super-hydration, under the subduction interface environment. Pyrophyllite usually occurs in low-grade metamorphosed Al-rich sediments and also in high-pressure/low-temperature metamorphic rocks[27]. Interestingly, it has been established that pyrophyllite is formed by the reaction between kaolinite and quartz in the pressure and temperature range of 0.1–0.2 GPa and 250–260 °C, respectively[28]. Previous works also reported that

pyrophyllite decomposes into the $Al_2SiO_5$-$SiO_2$ polymorph assemblage (andalusite/sillimanite/kyanite-quartz/coesite) at ~2.5–5.0 GPa and 500–900 °C ranges[29,30]. Pyrophyllite has, however, not yet been investigated under subduction conditions as a reaction product of kaolinite (and quartz) and hence the combined impact of subducting clays in deep water transportation has been unknown. In this work, by using the combination of a resistively-heated diamond-anvil cell (RH-DAC) and in-situ synchrotron X-ray powder diffraction (XRD), we have investigated the stability of pyrophyllite up to ~23 GPa and ~900 °C corresponding to 600–700 km depth range, following cold subduction thermal models, i.e., in a range of cold slab surface geotherms from South Mariana and Kermadec subduction with <5 °C/km to Izu-Bonin subduction with ~6 °C/km models[31–35]. Typically, cold slab geotherms could represent ~28.5% of the global subduction system[35,36], and the P-T conditions applied in this study can account for approximately half of the cold slabs in the contemporary Earth. We observed sequential breakdowns of pyrophyllite involving the formation of further hydrated minerals and stable hydrous phases in the ASH system, to deliver ~5.0 wt% of water across the mantle transition zone (MTZ) down to the lower mantle.

## Results and discussion
### Hydration breakdown of pyrophyllite along cold subduction geotherm
In-situ HP-HT synchrotron XRD experiments on pyrophyllite were performed up to ~23 GPa and 700–900 °C following the cold subduction geotherms of the South Mariana, Kermadec, and Izu-Bonin thermal models[35] (Figs. 1 and 2, Supplementary Table 1, and Supplementary Figs. 1–8). In water-rich cold subduction conditions, pyrophyllite transforms into gibbsite $(Al(OH)_3)$ and diaspore $(AlO(OH))$ above 4.4(4) GPa and 405 ± 60 °C, i.e., ~135 km depth, by releasing the silica component as coesite. The co-formation of these hydrated minerals can be explained by their close relative enthalpies of formation at the corresponding pressure range (Supplementary Text 1 and Supplementary Fig. 9a). From a crystal-chemical point of view, both hydrous phases form by the removal of the silicon tetrahedral sheets and then different degrees of hydration and connectivity of the remaining aluminum octahedral sheets (Fig. 2 and Supplementary Text 2). Upon further pressure and temperature increase above 5.9(5) GPa and 495 ± 70 °C, equivalent to ~185 km depth conditions, selective dehydration of gibbsite initiates resulting in the formation of a diaspore + coesite assemblage (Fig. 1). This is also in line with the calculated enthalpies where a crossover in the relative stabilities between gibbsite and diaspore occurs near 5.0 GPa (Supplementary Text 1 and Supplementary Fig. 9a). Under moderately cold slab surface geotherm, i.e., following the thermal model of Izu-Bonin subduction[35], for which data have been independently measured and analyzed, pyrophyllite showed the breakdown into diaspore + coesite assemblage above 4.0(2) GPa and 535 ± 40 °C, i.e., ~125 km depth (Fig. 1), without undergoing a phase assemblage region having gibbsite. Under these relatively low thermal conditions, the actual depths for phase change could be altered when reaction kinetics are considered, together with other chemical/phase components existing in a real subducting layer.

Above 7.4(5) GPa and 550 ± 80 °C, i.e., ~230 km depth, topaz $(Al_2SiO_4(OH)_2)$ starts to appear in the diaspore + coesite assemblage[37,38] (Fig. 1). However, if the temperature increases beyond the diaspore + coesite stability region to >900 ± 50 °C near 6.3(4) GPa, mimicking the stagnation of the cold subducting slab near 195 km depth, dehydration of diaspore is induced resulting in the formation of a different phase assemblage containing kyanite $(Al_2SiO_5)$ (Fig. 1)[37]. On the other hand, when the cold slab continues to subduct without being stagnated below 255 km depth, coesite turns to stishovite and forms a phase assembly of topaz + diaspore. Stishovite formed in our experiments is considered to be anhydrous as recent experimental studies have shown that anhydrous stishovite is preferred above 600 °C[39–41].

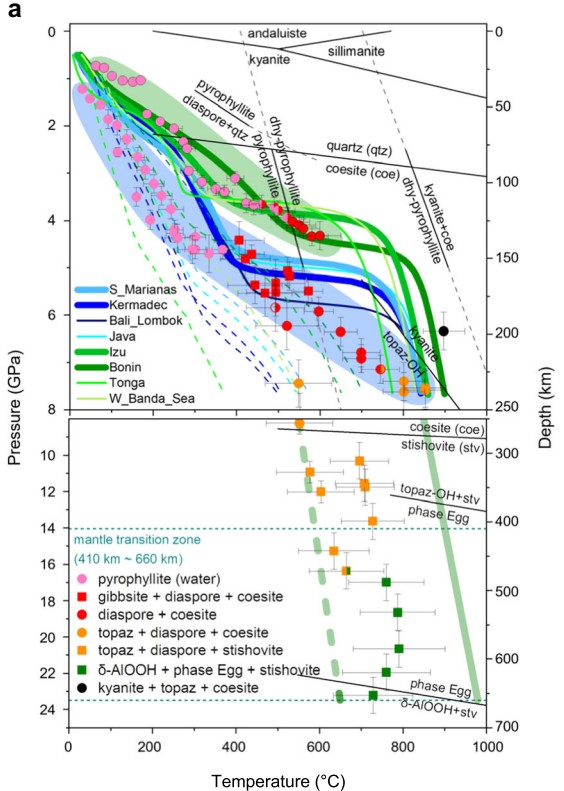

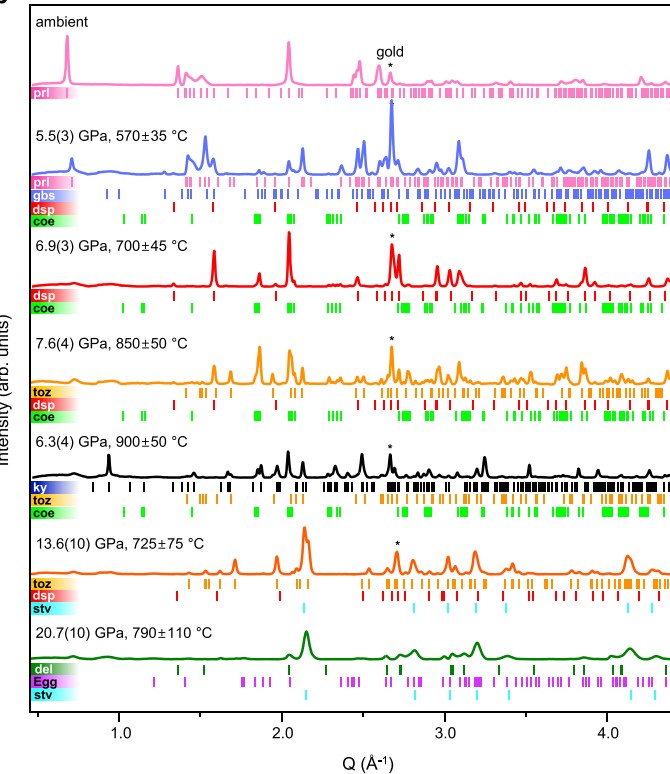

**Fig. 1 | Experimental P-T conditions and in-situ X-ray diffraction patterns of pyrophyllite and its breakdown products along water-rich cold subduction geotherms. a** The stability of pyrophyllite and its decomposition products at P-T conditions of cold slab surface geotherms (blue and green bands denote the regions of P-T conditions along the coldest and moderately cold slab surface geotherms, respectively, from W1300 model by Syracuse et al. (2010)). The error bars represent pressure and temperature uncertainty (see Supplementary Table 1). Continuous and dashed curves (upper panel) denote the geotherms of the sub-ducting slab surfaces and corresponding slab Moho, respectively[35]. The geotherm models within the mantle transition zone (lower panel) have been extrapolated from the data by refs. 93,94. Straight black lines represent previously established phase boundaries of the aluminous phases and SiO₂ polymorphs[27,29,37,95–99]. The horizontal dotted lines define the upper and lower boundaries of the mantle transition zone. **b** Representative X-ray powder diffraction (XRD) pattern of pyr-ophyllite and its products. The backgrounds of the XRD patterns have been sub-tracted prior to present (see Supplementary Figs. 1–7). Sequential formations of gibbsite, diaspore, topaz, kyanite, phase Egg, and δ-AlOOH are observed along the hydrous cold subduction geotherm, together with SiO₂ polymorph (coesite or stishovite). *Phase abbreviations: pyrophyllite (prl), gibbsite (gbs), diaspore (dsp), coesite (coe), topaz (toz), kyanite (ky), stishovite (stv), phase Egg (egg), δ-AlOOH (del).

After the slab penetrates the upper part of the MTZ near 490 km depth, i.e., ~17(1) GPa and 760 ± 90 °C, a phase assemblage containing phase Egg (AlSiO₃(OH)) and δ-AlOOH are formed together with stishovite, which persists down to the bottom of the MTZ near 600–700 km depth conditions, i.e., ~21–23(1) GPa and 730–790 ± 110 °C (Fig. 1).

In order to simulate a more realistic natural subduction condi-tions, another set of in-situ HP-HT experiments was performed using a solution containing NaCl-MgCl₂ (Supplementary Table 1 and Supple-mentary Figs. 10 and 11). Numerous studies demonstrated that major and trace elements (or solutes) are dissolved in subduction zone fluids (SZFs) and recycled in the deep mantle[8,12,42–44]. In the SZFs, salts such as NaCl and MgCl₂ are identified as important components[42–44], and the salinity of such fluid ranges between 5 and 15 wt%[45,46]. Furthermore, Holland and Ballentine (2006) revealed that subduction of sediment and seawater-dominated pore fluids play an important role in under-standing the elemental abundance pattern of heavy noble gases (Ar, Kr, and Xe) in the mantle to be remarkably close to that of seawater[47], i.e., recycling of seawater is responsible for the observed heavy noble gas pattern in the convecting mantle as subduction occurs in a seawater-dominated fluid environment. Under such ternary seawater composition, breakdown into the diaspore + coesite assemblage was confirmed above 5.4(2) GPa and 415 ± 30 °C, i.e., ~170 km depth, which was followed by the formation of the topaz + coesite assemblages

above 7.1(2) GPa and 500 ± 40 °C, i.e., ~220 km depth, confirming the overall breakdown sequence of subducting pyrophyllite in pure water conditions. In contrast, when the experiment is performed under water-free condition using silicone oil as a pressure-transmitting medium (PTM) along the same cold subduction geotherm, pyr-ophyllite remained stable up to ~7.0(4) GPa and 585 ± 40 °C, i.e., ~220 km depth, without undergoing the observed hydration break-down under pure and saline water media (Supplementary Figs. 10 and 11).

## Changes in the net water contents of the phase assemblages in the ASH system

As the subducting plates reach the base of the upper mantle, some become stagnated while others penetrate, depending on the physical and chemical resistance encountered in the mantle transition zone (MTZ). Nearly ~70% of cold subducting slabs are found to interact with the MTZ as observed by the transition zone-slab morphologies from tomographic studies and their Benioff stress state[48,49]. In such a case, a subducting channel, which is defined as the topmost sedimentary layer with a thickness of up to ~2–3 km at the interface between the sub-ducting crust and the base of the upper mantle[7,50], becomes tectoni-cally distributed and metamorphically deformed in the MTZ[51,52]. In the upper mantle region, we observed that pyrophyllite in a water-rich cold subducting channel undergoes sequential breakdowns/

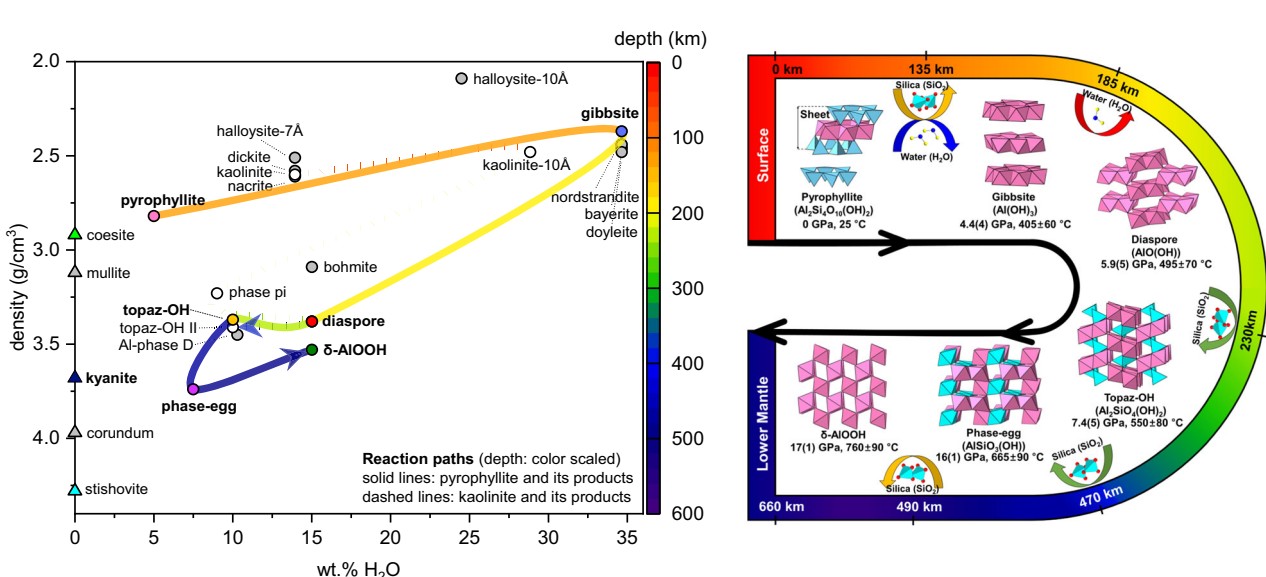

**Fig. 2 | Reaction paths and structural evolution of pyrophyllite into its breakdown products in the ASH system. a** The reaction path from pyrophyllite (solid line) is compared to that from kaolinite (dotted line) with their colors matching the formation depths. The reaction products from pyrophyllite and kaolinite are denoted by colored and white circles, respectively, while the other aluminosilicate minerals within the ASH system are marked with gray symbols. **b** The structural evolution from pyrophyllite is shown in the order of the formation depths to illustrate the incorporation or release of $H_2O/SiO_2$ components (Supplementary Text 2).

transformation to form gibbsite + diaspore + coesite near 135 km depth, diaspore + coesite near 185 km depth, and then topaz + diaspore + coesite assemblages near 230 km depth (Figs. 1 and 3a, and Table 1). In the form of structural hydroxyls (OH), pyrophyllite contains ~5.0 wt% $H_2O$, whereas gibbsite represents the most hydrated mantle mineral with ~34.6 wt% $H_2O$, thereby making the formation of the gibbsite + diaspore + coesite assemblage a hydration breakdown reaction, leading to a net water content increase by ~90%, i.e., an increase from net ~5.0 wt% to net ~9.5 wt% (or to net 26.1 wt% counting only the ASH phases, Fig. 4b and Table 1). This would remove a portion of subducting fluid by ~1.8 wt% near 135 km depth (Table 1 and 2). The subsequent disappearance of gibbsite and the formation of the diaspore + coesite assemblage indicates that a dehydration breakdown reaction occurred as the net water content decreased by ~47% back to ~5.0 wt% (Fig. 4b and Table 1). This would then release the same amount of fluid stored from the hydration breakdown near 135 km depth back into the subduction interface near 185 km depth. Throughout the region of the topaz + diaspore + coesite assemblage from ~230 km depth, the water content is preserved and net ~5.0 wt% $H_2O$ is transported down to the MTZ (Fig. 4b and Table 1). Under the moderately cold slab surface geotherms, the initial water content of ~5.0 wt% is preserved through the formation of the diaspore + coesite assemblage down to ~125–135 km depth.

When the subducting slab or channel bends within the upper mantle parallel to the MTZ, a phase assemblage containing anhydrous kyanite would be formed from the diaspore + coesite assemblage below ~195 km depth, which would constitute a dehydration breakdown reaction to release $H_2O$ by net ~2.7 wt% (Fig. 1 and Table 1). On the other hand, when the subducting slab penetrates through the MTZ, another water-preserving transformation would occur to form the δ-AlOOH + phase Egg + stishovite assemblage, which is stable to deliver the net ~5.0 wt% $H_2O$ towards the lower mantle (Fig. 4b and Table 1). This is in contrast to the previous results where pyrophyllite has been known to undergo facile dehydration under water-free condition to form kyanite + coesite assemblage near 5.0 GPa and 1200 °C[29]. We therefore demonstrate that the presence of water/fluid plays a dictating role in the stability of pyrophyllite and induces sequential phase transformations along the ASH system, transporting the initial amount of water as contained in the original pyrophyllite, i.e., ~5.0 wt% $H_2O$, down to the lower mantle in the form of the final phase assemblage of δ-AlOOH + phase Egg + stishovite (Fig. 4b and Table 1).

## Changes in the net crystalline density of the phase assemblages in the ASH system

It is also important to correlate the changes in the net water contents to the net crystalline density in the observed phase assemblages (Fig. 4b and Table 1). Throughout the observed breakdown sequence, the net crystalline density increases by a total of ~49.6% to compete with the density of the corresponding depth regions. During the hydration breakdown reaction, the net crystalline density increases discontinuously by ~5.3% from 2.82 g/cm³ in pyrophyllite to 2.97 g/cm³ in the gibbsite + diaspore + coesite assemblage. As gibbsite disappears with the dehydration breakdown into the diaspore + coesite assemblage, the net density increases further by ~6.1% to 3.15 g/cm³, which then decreases marginally to 3.14 g/cm³ upon the water-preserving breakdown to the topaz + diaspore + coesite assemblage. A significant increase in the net density by ~30.2% to 4.10 g/cm³ is then driven by the transformation of coesite to stishovite in the above assemblage. The net density of the topaz + diaspore + stishovite assemblage would then be ~11.3% higher than the average density of 3.72 g/cm³ below the 410 km discontinuity, allowing the assemblage to penetrate into the MTZ. Within the MTZ, the net density increases further by ~1.7% to 4.21 g/cm³ upon the formation of the δ-AlOOH + phase Egg + stishovite assemblage. The expected density of such an assemblage would then be higher by ~2.4% than the average density of 4.19 g/cm³ in the region below the 660 km discontinuity, enabling the subducting channel to be gravitationally favorable to drive its subduction into the lower mantle region.

It has long been known that either of the two representative petrological models of the MTZ, i.e., pyrolite[53] and piclogite models[54], cannot firmly explain the observed seismological $v_s$ models (PREM: Preliminary Reference Earth Model)[55,56]. With this regard, it has been

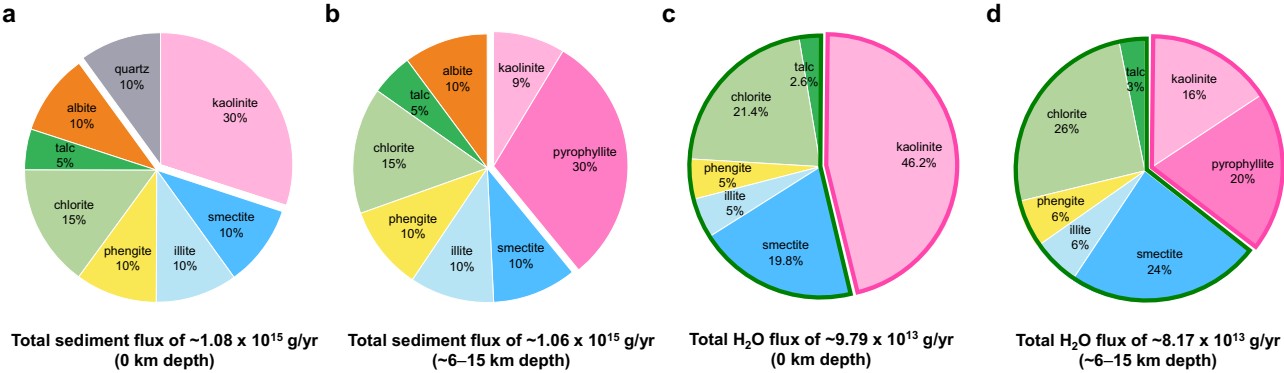

**Fig. 3 | Total sediment and H₂O fluxes of subducting sedimentary layer.** The phi charts show the average mineral makeup of oceanic sediment and its H₂O flux in a subducting slab (thick solid lines in pink for ASH system and green for non-ASH system) (**a**) at trench (0 km depth) and (**b**) after the formation of pyrophyllite (~6–15 km depth) (Table 2 and references therein).

**Table 1 | Calculated density and water contents of pyrophyllite and its breakdown products at different P-T conditions**

| Depth (km) | Pressure (GPa) | Temperature (°C) | Phase assemblages | Phases | Density (g/cm³) | Net crystalline density (g/cm³) | Net water contents with SiO₂ (wt% H₂O) | Net water contents without SiO₂ | Net crystalline and fluid volume (ratio, cm³) |
|---|---|---|---|---|---|---|---|---|---|
| 0 | 0 | 25 | prl | pyrophyllite | 2.82(1) | 2.82 | 5.0 | 5.0 | 1000:1000 |
| ~170 | 5.5(3) | 570 ± 35 | gbs + dsp + 4 coe | gibbsite | 2.37(2) | 2.97 | 9.5 | 26.1 | 1020:859 |
| | | | | diaspore | 3.37(1) | | | | |
| | | | | coesite | 3.01(1) | | | | |
| ~215 | 6.9(3) | 700 ± 45 | 2 dsp + 4 coe | diaspore | 3.41(1) | 3.15 | 5.0 | 15.0 | 897:1000 |
| | | | | coesite | 3.02(1) | | | | |
| ~240 | 7.6(4) | 850 ± 50 | 1/2 toz + dsp + 7/2 coe | topaz | 3.40(1) | 3.14 | 5.0 | 12.0 | 887:1000 |
| | | | | diaspore | 3.41(1) | | | | |
| | | | | coesite | 3.03(1) | | | | |
| ~400 | 13.6(10) | 725 ± 75 | 1/2 toz + dsp + 7/2 stv | topaz | 3.56(1) | 4.10 | 5.0 | 12.0 | 710:1000 |
| | | | | diaspore | 3.51(1) | | | | |
| | | | | stishovite | 4.34(1) | | | | |
| ~585 | 20.7(10) | 790 ± 110 | del + egg + 3 stv | δ-AlOOH | 3.82(1) | 4.21 | 5.0 | 10.0 | 676:1000 |
| | | | | phase Egg | 4.00(1) | | | | |
| | | | | stishovite | 4.41(1) | | | | |
| ~670 | 24.0(10)* | 790 ± 110 | del + egg + 3 stv | δ-AlOOH | 3.89(2) | 4.29 | 5.0 | 10.0 | 663:1000 |
| | | | | phase Egg | 4.06(2) | | | | |
| | | | | stishovite | 4.50(4) | | | | |
| ~195 | 6.3(4) | 900 ± 50 | 1/2 ky + 1/2 toz + 3 coe (ky + 3coe) | kyanite | 3.70(1) | 3.41 | 2.3 | 5.3-10.0 | 846:1070 (811:1141) |
| | | | | topaz | 3.42(1) | | | | |
| | | | | coesite | 3.00(1) | | | | |

Density of each phase was calculated using ρ = M V Z⁻¹, where M is the molecular weight, Z is the number of the formula unit per unit cell, and V is the volume derived from the profile fitting of XRD data using GSAS Program (Supplementary Figs. 1–7). Net crystalline density was calculated based on the proportions (and the equations of state) of composing mineral phases (marked with an asterisk).

suggested that the role of a subducting slab may account for the anomalous seismic wave velocity and density changes within the region[51,56,57]. If the subduction of a sedimentary layer continues down to the MTZ, hydrous mineral assemblages in the ASH system, as observed in our study, could play an important role in impacting the physicochemical characteristics of the region, at least on a regional scale, and account in part the observed seismic anomalies. Our experimental results should be further examined to identify its potential relationship to earthquake nucleation as well as seismic anomalies along the subducting channel down to the MTZ region.

### New estimate on the H₂O flux from subducting clays in the ASH system

While the H₂O flux from hydrous minerals in the subducting crust and upper mantle has been estimated down to ~150–230 km depth range to be in the range of ~1.98–3.16 × 10¹⁴ g/yr (ref. 2), the net H₂O contribution from the subducting sedimentary layer has so far been underestimated or overlooked. Based on the average mineral makeup of oceanic sediments (Tables 2 and 3), the initial H₂O flux by sediments can be estimated to be ~9.79 × 10¹³ g/yr (Fig. 3a). By an assumption that pyrophyllite would be formed from subducting kaolinite and quartz[28] and these clay minerals undergo (super-) hydration breakdowns as observed in this and previous studies[26] along the coldest thermal models, we can estimate the changes in the average H₂O flux of a subducting sedimentary layer down to ~185 km depth (Table 2). By the depth of ~6–15 km, kaolinite and quartz, contained in the initial sedimentary layer by ~30 and ~10 wt%, respectively, would react to form pyrophyllite to account ~30 wt% of the sedimentary layer (Fig. 3). This would change the H₂O flux by sediments to ~8.17 × 10¹³ g/yr (Table 3 and Fig.3b). Super-hydration of the remaining kaolinite (~9 wt%) would occur at ~75 km depth to increase the H₂O flux of sediments by ~24% (net ~7% increase to ~8.75 × 10¹³ g/yr when considering dehydration

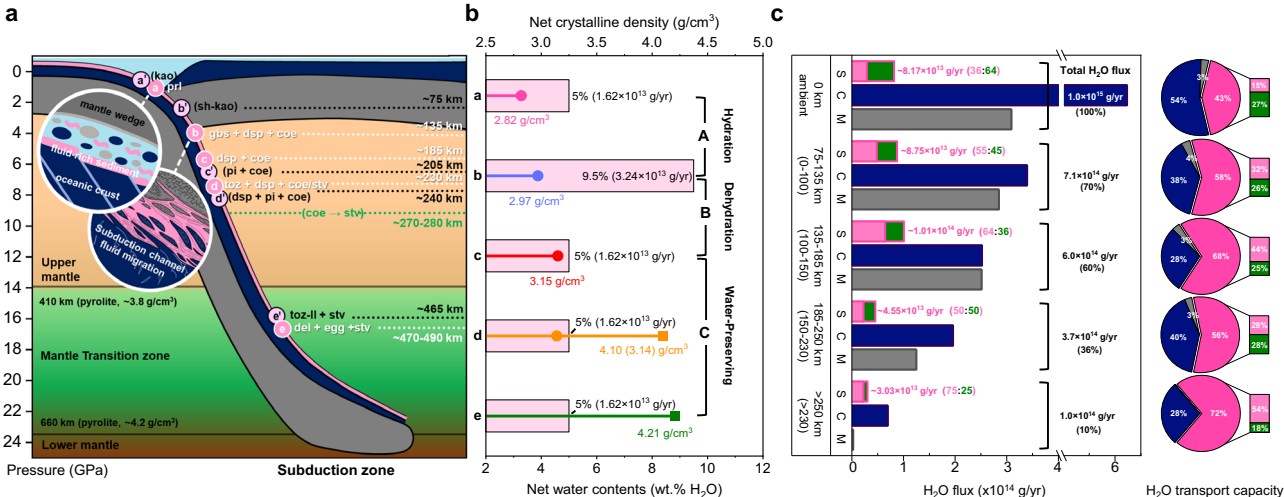

**Fig. 4 | Sequential breakdowns from pyrophyllite (and kaolinite) and its impact on H₂O flux along subduction zone. a** Depths of sequential breakdowns from pyrophyllite (and kaolinite) are marked in a schematic subduction diagram. Insets are the schematic illustrations of the fluid migration pathways in shallow and intermediate subduction interfaces (modified after Konrad-Schmolke et al. (2011)). The phase assemblages from pyrophyllite are: a (prl), b (gbs + dsp + coe), c (dsp + coe), d (toz + dsp + coe/stv), e (del + Egg + stv), and from kaolinite: a′ (kao), b′ (sh-kao), c′ (pi + coe), d′ (dsp + pi + coe), e′ (toz-II + stv) (Supplementary Table 5). *Phase abbreviations: pyrophyllite (prl), gibbsite (gbs), diaspore (dsp), coesite (coe), topaz (toz), stishovite (stv), phase Egg (egg), δ-AlOOH (del) and kaolinite (kao), super-hydrated kaolinite (sh-kao), phase-pi (pi), topaz-II (toz-II). **b** Estimated net water contents (wt% H₂O) and net crystalline density (g/cm³) of the respective phase assemblages from pyrophyllite. Values in the parenthesis are the estimated H₂O flux from pyrophyllite. **c** H₂O flux of a subducting slab composed of sediments (S, pink boxes; filled in pink for ASH system and filled in green for non-ASH system), crust (C, navy boxes), and hydrated mantle (M, gray boxes) layers. Depth ranges for C and M down to <230 km are shown in parenthesis. The summed H₂O flux of S, C, and M layers at the surface is set to 100% to show the efficiency of H₂O transportation at depths. The pie charts in the right panel show H₂O transport capacity of each layer per unit volume.

reactions of other minerals composing sediments, see Supplementary Text 3 and Table 3), which will increase further by ~19% (net ~16% increase to ~1.01 × 10¹⁴ g/yr, see Supplementary Text 3 and Table 3) via the hydration breakdown of pyrophyllite at ~135 km depth (Tables 2 and 3). While super-hydrated kaolinite would remain stable down to ~205 km depth, the dehydration breakdown of the phase assemblage formed from pyrophyllite would occur at ~185 km depth and reduce the H₂O flux of sediments by ~19% to ~6.75 × 10¹³ g/yr (Tables 2 and 3). Therefore, by the depth of ~185 km, the H₂O flux of a subducting sedimentary layer increases in steps and becomes ~8.37 × 10¹³ g/yr (maximum ~1.01 × 10¹⁴ g/yr between ~135 and 155 km, see Supplementary Text 3 and Table 3) where contribution from clay minerals in the ASH system, i.e., kaolinite and pyrophyllite, becomes ~64% (Fig. 4 and Tables 2 and 3). Such an amount is ~2/3 of the total water transport capacity by a subducting sedimentary layer, which would then account ~40% of the average water transport capacity by a subducting crust. At depths below ~185 km, breakdowns would occur from both super-hydrated kaolinite and the phase assemblage from pyrophyllite to form assemblages containing diaspore and topaz to deliver ~1.62–2.26 × 10¹³ g/yr H₂O (Fig. 4 and Tables 2 and 3). This H₂O flux would remain preserved through the MTZ region down to the lower mantle whereas dehydration of other hydrous minerals in the sediment is completed by ~350 km (equivalent to ~11 GPa conditions in Table 3).

While the total H₂O flux in a subducting slab decreases down to ~60% by the depth of ~185 km (or ~150 km in the case of subducting crust and upper mantle), the H₂O flux by the ASH system. i.e., kaolinite, pyrophyllite, and their derivatives, increase by ~122% due to the hydration breakdown of pyrophyllite and super-hydration of kaolinite (Table 2 and Fig. 4c). By the depth of ~250 km (or ~230 km in the case of subducting crust and upper mantle), the total H₂O flux of a subducting slab is reduced to ~10% by dehydration (breakdown) of minerals within respective layers. The H₂O flux of the composing layers, however, reveals the increasing role of a sedimentary layer as its H₂O transport capacity per unit volume increases from ~43% to ~72% while that of a crust decreases from ~54% to ~28% (Fig. 4c and Tables 2 and 3). Within

the sedimentary layer, the H₂O flux also reveals the increasing role by the minerals in the ASH system as its H₂O transport capacity per unit volume increases from ~15% to ~54% while that by non-ASH minerals decreases from ~27% to ~18% by the depth of ~250 km.

Modern-style subduction, as characterized by the formation of blueschist facies, has been operating since Paleoproterozoic (~2.2–2.0 Ga)[58]. We may infer then about 2.3–2.5% of the current ocean mass could have been transported by the subducting clays in the ASH system into the lower mantle (Tables 2 and 3). Further studies will be necessary to fully understand the role of the subducting sedimentary layer on the global H₂O flux into the interior of the Earth as several uncertainties remain pertaining to the possibly overlooked and unknown hydration and dehydration processes of other mineral phases and their geochemical and geodynamical behaviors under diverse subduction environments.

## Methods

### Sample and initial characterization
Initial characterization of the pyrophyllite (Nowhado, South Korea, $Al_2Si_4O_{10}(OH)_2$) sample used in this study was performed using synchrotron X-ray diffraction (XRD) and field emission scanning electron microscopy (FE-SEM, JEOL-7800F) equipped with energy-dispersive spectrometer (EDS, Oxford Instruments) (Supplementary Fig. 1). Two-dimensional intensity data from synchrotron XRD were converted into one-dimensional data using the Dioptas Program[59], which were then analyzed using the Le Bail method[60] implemented in the GSAS suite of programs[61]. Pyrophyllite at ambient conditions was indexed with the space group C1̄ resulting in refined cell parameters of a = 5.1506(5) Å, b = 8.9468(11) Å, c = 9.3792(6) Å, α = 92.37(2)°, β = 100.13(2)°, γ = 88.56(1)°.

### In-situ high-pressure and high-temperature synchrotron X-ray powder diffraction
In-situ synchrotron X-ray powder diffraction measurements were performed at the Extreme Conditions Beamline (ECB) P02.2 at PETRA-

**Table 2 | Estimated global $H_2O$ flux by subducting sediments and crust (with mantle)**

| Parameters | Value | References |
|---|---|---|
| thickness of oceanic crust | 5–8 km (av. 7 km) | Geissler et al. (2017) and White et al. (1992)[72,73] |
| volume of oceanic crust | $2100 \times 10^6$ km³ | Wyllie (1971)[74] |
| mass of oceanic crust | $6.07 \times 10^{21}$ kg | Ronov and Yaroshevsky (1969)[75] |
| thickness of oceanic sediment | av. 404–927 m | Straume et al. (2019) and references therein[76] |
| volume of oceanic sediment | $113\text{-}337 \times 10^6$ km³ | Hay (1988) and references therein[14] |
| mass of oceanic sediment | $0.2\text{-}0.4 \times 10^{21}$ kg | |
| mean depth of ocean | av. 3.7 km | Charette and Smith (2010)[77] |
| volume of ocean | $1332 \times 10^6$ km³ | Mackenzie and Garrels[78] |
| mass of ocean | $1.4 \times 10^{21}$ kg | |
| total amount of sediments in trenches | | |
| sediment fluxes | $1.43 \times 10^{15}$ g/yr | Rea and Ruff (1996)[4] |
| (terrigenous sediment) | $(1.08 \times 10^{15}$ g/yr$)$ | |
| water fluxes | $0.91 \times 10^{15}$ g/yr | |
| terrigenous sediment fluxes | | Calculated data based on |
| kaolinite (30%) | $3.24 \times 10^{14}$ g/yr | Leinen (1989), Li and Schoonmaker (2003), Windom (1976) and refer- |
| quartz (10%) | $1.08 \times 10^{14}$ g/yr | ences therein[79–81] |
| smectite-illite (20%), phengite (10%), chlorite (15%), talc (5%), plagioclase (10%) | $6.48 \times 10^{14}$ g/yr | |
| **$H_2O$ flux by clays and their super-hydration/breakdown products in the ASH system** | | |
| subducting kaolinite and quartz[a] | $\sim 2.90 \times 10^{13}$ g/yr | Calculated data based on |
| pyrophyllite (form at ~10 km depth) | $\sim 1.62 \times 10^{13}$ g/yr | Matsuda et al. (1992), Leinen (1989), Li and Schoonmaker (2003), |
| (remaining) kaolinite | $\sim 1.28 \times 10^{13}$ g/yr | Windom (1976) and references therein[28,79–81] |
| (super-) hydration breakdown by ~185 km depth[b] | $\sim 6.45 \times 10^{13}$ g/yr | |
| gibbsite + diaspore + coesite | $\sim 3.24 \times 10^{13}$ g/yr | |
| super-hydrated kaolinite | $\sim 3.21 \times 10^{13}$ g/yr | |
| dehydration breakdown by ~250 km[c] | $\sim 2.26 \times 10^{13}$ g/yr | |
| topaz + diaspore + coesite | $\sim 1.62 \times 10^{13}$ g/yr | Estimated based on this study and Hwang et al. (2017)[26]. |
| (kyanite + diaspore + coesite) | $( < 0.81 \times 10^{13}$ g/yr$)$ | |
| phase-pi + diaspore + coesite | $\sim 0.64 \times 10^{13}$ g/yr | |
| water-preserving breakdown in the MTZ region[d] | $\sim 1.62\text{-}2.26 \times 10^{13}$ g/yr | |
| δ-AlOOH + phase Egg + stishovite | $\sim 1.62 \times 10^{13}$ g/yr | |
| topaz-II + stishovite | $< 0.64 \times 10^{13}$ g/yr | |
| **Total $H_2O$ flux by subducting slab** | | |
| subducting terrigenous sediments (without ASH system) | | Estimated based on |
| 100 km depth | $\sim 5.27 \times 10^{13}$ g/yr | previous studies (details in Table 3) |
| 100–150 km depth | $\sim 3.62\text{-}3.93 \times 10^{13}$ g/yr | |
| 150–250 km depth | $\sim 1.93\text{-}2.29 \times 10^{13}$ g/yr | |
| >250 km depth | $< 0.76\text{-}1.78 \times 10^{13}$ g/yr | |
| subducting crust (and mantle) | | |
| 100 km depth | $\sim 3.42\text{-}6.32 \times 10^{14}$ g/yr | Calculated data based on |
| 100–150 km depth | $\sim 2.52\text{-}5.02 \times 10^{14}$ g/yr | van keken et al. (2011) and references therein[2] |
| 150–230 km depth | $\sim 1.98\text{-}3.16 \times 10^{14}$ g/yr | |
| >230 km depth | $< 0.69\text{-}0.72 \times 10^{14}$ g/yr | |

[a]Kaolinite and quartz react to form pyrophyllite by the reaction: kaolinite + 2quartz = pyrophyllite + water (ref. 28).

[b]Pyrophyllite and water reacts to form the gibbsite + diaspore + coesite assemblage (this study), while kaolinite and water reacts to form super-hydrated kaolinite[26].

[c]topaz + diaspore + coesite assemblage is formed in the upper mantle region along the breakdown sequence from pyrophyllite (this study), while phase-pi + diaspore + coesite is formed along that of super-hydrated kaolinite[26].

[d]δ-AlOOH + phase Egg + stishovite assemblage is formed in the MTZ region along the breakdown sequence from pyrophyllite (this study), while topaz-II + stishovite assemblage is formed along that of super-hydrated kaolinite[26].

III, Germany, 3D-XRS and 6D beamlines at PLS-II, South Korea, and 13-BMC beamline and 16-BMD beamline at APS, USA. At beamline P02.2, the X-ray beam from the undulator source was tuned to a wavelength of 0.4834(1) Å (25.650 keV) and focused to $8 \times 4$ μm² in size (FWHM) using Compound Refractive Lense (CRL) optics. A Perkin Elmer XRD 1621 detector was used to collect diffraction data at distance of ~401 mm from the sample with 10 sec of exposure time. At beamline 3D, the X-ray beam from the bending magnet source was tuned to a wavelength of 0.6886(1) Å (18.005 keV) and focused to 100 μm in size using a double crystal monochromator of bent Si(111) and Si(311) crystals. A Mar345 imaging plate detector was used to collect diffraction data at distance of ~310 mm from the sample with 120 sec of exposure time. At beamline 6D, the X-ray beam from the bending magnet was tuned to a wavelength of 0.6530(1) Å (18.986 keV) and focused to 100 μm in size using a double crystal monochromator and a toroidal mirror. A 2D CCD detector (MX225-HS, Rayonix L.L.C., USA)

was used to collect diffraction data at distance of ~253 mm with 60 sec of exposure time. At beamline 13-BMC, the X-ray beam from the bending magnet source was tuned to a wavelength of 0.4340(1) Å (28.568 keV) and focused to $12 \times 18$ μm² in size using a KB-mirror. A PILATUS 1 M detector was used to collect diffraction data at a distance of ~169 mm from the sample with 150 sec of exposure time. At beamline 16-BMD, the X-ray beam from the bending magnet source was tuned to a wavelength of 0.4959(1) Å (25.000 keV) and focused to $4 \times 4$ μm² in size (FWHM) using a KB-mirror. A PILATUS 1 M detector was used to collect diffraction data at distance of ~250 mm from the sample with 150 sec of exposure time. With a small beam size at DESY and APS, we collected the data with sample movement ($3 \times 3$ or $5 \times 5$ grid measurements) at intervals of ~10 μm. As a high-pressure vessel, a symmetric-type diamond-anvil cell (DAC) equipped with a pair of type-I anvils of culet diameter of 300 μm or 500 μm was used in combination with a membrane device for online pressure control. A rhenium gasket

**Table 3 | Stability of representative hydrous phases in the subducting sediments along cold subduction conditions**

| P (GPa) | kao | prl | tlc | chl | smec | ill | phg | ab | H$_2$O flux (g/yr) |
|---|---|---|---|---|---|---|---|---|---|
| 0.5 | | | | | | | | | 8.17 x 10$^{13}$ |
| 1 | | | | | | | | | 8.17 x 10$^{13}$ |
| 1.5 | a' | | | | smec 2W | | | ab | 8.17 x 10$^{13}$ |
| 2 | | | tlc | | | ill | | | 8.17 x 10$^{13}$ |
| 2.5 | | a | | | | | | | 8.17 x 10$^{13}$ |
| 3 | | | | chl | | | | | 8.75 x 10$^{13}$ |
| 3.5 | | | | | | ill | | smec 1W | 8.75 x 10$^{13}$ |
| 4 | | ~135 km | | | | | | | 8.74 x 10$^{13}$ |
| 4.5 | b' | | | | | | | jd | 1.01 x 10$^{14}$ |
| 5 | | b | | | | | phg | | 1.01 x 10$^{14}$ |
| 5.5 | | ~185 km | | | | | | | 8.37 x 10$^{13}$ |
| 6 | | | 10Å | HAPY | | | | | 6.75 x 10$^{13}$ |
| 6.5 | c' | c | | | msc | msc | | | 4.19 x 10$^{13}$ |
| 7 | | ~230 km | | | | | | | 4.56 x 10$^{13}$ |
| 7.5 | | | | | | | | | 4.56 x 10$^{13}$ |
| 8 | | | | 23Å | | | | | 4.56 x 10$^{13}$ |
| 9 | | | ens | | | | | | 4.04 x 10$^{13}$ |
| 10 | | | | A | | | | | 3.44 x 10$^{13}$ |
| 11 | d' | d | | E | | | K-hld+toz | | 3.03 x 10$^{13}$ |
| 12 | | | | | | | | | 2.26 x 10$^{13}$ |
| 13 | | | | | | | | | 2.26 x 10$^{13}$ |
| 14 | | | | | | | | | 2.26 x 10$^{13}$ |
| 15 | | ~470 km | | | | | | | 2.26 x 10$^{13}$ |
| 16 | e' | | | | | | | | 2.26 x 10$^{13}$ |
| 17 | | | | | | | | | 1.62 x 10$^{13}$ |
| 18 | | | | | | | | | 1.62 x 10$^{13}$ |
| 19 | | | | | | | | | 1.62 x 10$^{13}$ |
| 20 | | e | | | | | | | 1.62 x 10$^{13}$ |
| 21 | | | | | | | | | 1.62 x 10$^{13}$ |
| 22 | | | | | | | | | 1.62 x 10$^{13}$ |
| 23 | | | | | | | | | 1.62 x 10$^{13}$ |
| 24 | | ~660 km | | | | | | | 1.62 x 10$^{13}$ |

Within hydrous phases, (super-) hydration reaction is represented by double lines; dehydration reaction by dashed line; and water-preserving reaction by single line.

Phase assemblages and abbreviations are in Table 1 and Supplementary Tables 5 and 6.

References are as follows: kaolinite[26] (Hwang et al. (2017)), pyrophyllite (in this study), talc[82,83] (Yamamoto & Akimoto (1977); Chinnery et al. (1999)), chlorite[84,85] (Gemmi et al. (2011); Cai et al. (2019) and reference therein), smectite-illite[86–88] (Van de Kamp (2008); Carniel et al. (2014); Stefani et al. (2014)), muscovite-phengite[89–92] (Schmidt (1996); Ono (1998); Domanik & Holloway (1996, 2000)), albite[71] (Hwang et al. (2021)).

of 250 μm in thickness was indented to 40–60 μm, and a hole of 150 μm or 200 μm in diameter was drilled in the center as a sample chamber using an electric discharge machine (EDM)[62].

Simultaneous P-T condition was created by using external resistive electrical heaters (a graphite foil heater (at PETRA-III) and a coil heater (at PLS and APS) surrounding the diamond anvils). The RH-DAC setup offers the advantage of providing homogeneous and stable temperature across the entire sample chamber[63]. During the experiment, we followed the P-T conditions proposed for the South Mariana-Kermadec subduction and Izu-Bonin subduction models to simulate the coldest and moderately cold subduction geotherms, respectively[35]. Temperature was monitored using a R-type or K-type thermocouple attached to the pavilion of the diamond anvil close to the sample with the maximum uncertainties of ±3 °C. The overall uncertainty in the sample temperature has been estimated by the differences between the recorded thermocouple temperature and the calculated temperature based on the unit-cell volume and the known thermal expansion coefficient of standard materials[63]. The pressure was calculated using the equation of state of Au pressure marker[64,65] included in the sample chamber. More detailed description of this type of RH-DAC setup can be found in the literature[63,66]. Initially, we compressed the sample to ~1.0–2.0 GPa and then increased the pressure and temperature in increments of ~0.5 GPa and 50 °C up to ~23 GPa and ~900 °C, respectively. During our RH-DAC experiments, the samples were heated for about 12 h (or more) to reach target temperature, where the sample temperature was kept for at least ~20–30 min to ensure no further changes are occurring (Supplementary Table 1).

**Computational calculations**

The relative forming enthalpies of the mineral phases observed in our experiments were calculated using first-principles method

(Supplementary Table 2). We performed density function theory (DFT) calculations implemented in Vienna Ab Inito Simulaton Package (VASP)[67]. We used atomic potentials generated based on a projector augmented-wave method (PAW)[68] and Generalized Gradient Approximations (GGA). In our calculations, a plane wave cutoff energy for the wave function was set to 1000 eV. Geometry optimizations were performed using conjugate gradients minimization until all the forces acting on the ions were less than 0.01 eV/Å per atom. K-point mesh with a spacing of about 0.03 Å$^{-1}$ was adopted. Structural relaxations were performed at various constant volumes, and the calculated energy-volume data at zero Kelvin were fitted to a third-order Birch-Murnaghan equation of state (EOS)[69]:

$$E(V) = E_0 + \frac{9V_0 B_0}{16}\left\{\left[\left(\frac{V_0}{V}\right)^{\frac{2}{3}} - 1\right]^3 B_0' + \left[\left(\frac{V_0}{V}\right)^{\frac{2}{3}} - 1\right]^2 \left[6 - 4\left(\frac{V_0}{V}\right)^{\frac{2}{3}}\right]\right\}, \quad (1)$$

where $E_0$ denotes the intrinsic energy at zero pressure, $V_0$ is the volume at zero pressure, $B_0$ is the bulk modulus, and $B_0'$ is the first pressure derivative of the bulk modulus. The fitted parameters at zero kelvin are summarized in Supplementary Table 2. The relationship between the pressure and volume at zero kelvin can be expressed as:

$$P(V) = \frac{3B_0}{2}\left[\left(\frac{V_0}{V}\right)^{\frac{7}{3}} - \left(\frac{V_0}{V}\right)^{\frac{5}{3}}\right]\left\{1 + \frac{3}{4}(B_0' - 4)\left[\left(\frac{V_0}{V}\right)^{\frac{2}{3}} - 1\right]\right\} \quad (2)$$

The enthalpies of the starting sample, its breakdown products, and the hydrous pressure medium, i.e., pyrophyllite ($Al_2Si_4O_{10}(OH)_2$), gibbsite ($Al(OH)_3$), diaspore ($AlO(OH)$), topaz ($Al_2SiO_4(OH)_2$), kyanite ($Al_2SiO_5$), phase Egg ($AlSiO_3(OH)$), $\delta$-$AlO(OH)$, coesite-stishovite ($SiO_2$) and ice VII, respectively, at pressures in the range of 0–20 GPa were calculated using the equation H = E + PV. Subsequently, the stabilities of these minerals were evaluated based on the relative enthalpies in the following reactions (Supplementary Fig. 9):

$$Al_2Si_4O_{10}(OH)_2 + 2H_2O \rightarrow 2Al(OH)_3 + 4SiO_2 \quad (3)$$

$$Al_2Si_4O_{10}(OH)_2 \rightarrow 2AlO(OH) + 4SiO_2 \quad (4)$$

$$Al_2Si_4O_{10}(OH)_2 \rightarrow Al_2SiO_4(OH)_2 + 3SiO_2 \quad (5)$$

$$Al_2Si_4O_{10}(OH)_2 \rightarrow Al_2SiO_5 + 3SiO_2 + H_2O \quad (6)$$

$$Al_2Si_4O_{10}(OH)_2 \rightarrow 2AlSiO_3(OH) + 2SiO_2 \quad (7)$$

$$Al_2Si_4O_{10}(OH)_2 \rightarrow 2\delta - AlO(OH) + 4SiO_2 \quad (8)$$

In addition, we have performed computational structure search to locate the hydrogen positions in pyrophyllite because there is no experimental data for the crystal structure of pyrophyllite that includes the location of hydrogen atoms. We have fixed experimentally driven atomic positions of Al, Si, and O atoms as well as the lattice constants and generated hydrogen positions randomly within the unit cell, adopting the method implemented in Ab Initio Random Structure Searching (AIRSS)[70] module. Amongst ~200 predicted crystal structures, we have identified one that maintains a well-stabilized crystal structure based on the lowest total energy (Supplementary Table 4).

**Net crystalline density and net fluid volume calculations**[71]
The density of each phase was calculated using $\rho_{calc}$ (g cm$^{-3}$) = (M Z V$^{-1}$) × (avogadro's number)$^{-1}$, where M is the molecular weight, Z is the

number of the formula unit per unit cell, and V (Å$^3$) is the volume derived from the profile fitting of XRD data using GSAS program (Supplementary Figs. 1–7). Net crystalline density accounts the proportions of the composing crystalline phases. Net crystalline density is described here as:

$$\rho Net\ crystalline\ density = a\rho_a + b\rho_b + \cdots + z\rho_z \quad (9)$$

where $a$, $b$, and $z$ mean the proportions of the phase assemblages (Table 1 and Supplementary Table 5). The calculated density of each phase, $\rho_a$, is derived from XRD data.

To determine the changes in the net fluid volume after the reaction, the initial volume of pyrophyllite and water per mole is defined as:

$$\frac{avogadro's\ number\left(\frac{atom}{mol}\right) \times unit\ cell\ volume\ of\ albite\,(cm^3)}{the\ number\ of\ the\ formula\ unit\ per\ unit\ cell\,(atom)} \quad (10)$$

Based on the phase assemblages (Table 1 and Supplementary Table 5), changes in the net fluid volume are then calculated.

## Data availability
All data generated or analyzed during this study are included in this article and its Supplementary Information. Source data is deposited in the figshare repository (https://doi.org/10.6084/m9.figshare.25611309). Any additional data are available from the corresponding author upon request.

## Code availability
The Vienna Ab initio Simulation Package is proprietary software available for purchase at https://www.vasp.at/. The other code for this study is available from the corresponding author upon request.

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

## Acknowledgements

This work was supported by the Leader Researcher program (NRF-2018R1A3B1052042) of the Korean Ministry of Science and ICT (MSIT). H.H. was supported by the National Research Foundation of Korea grant funded by the MSIT (RS-2022-001658). D.Y.K. and Y.H. was supported by the National Natural Science Foundation of China (U1930401). Synchrotron experiments were performed at ECB P02.2 at PETRA-III, 3D and 6D beamlines at PLS-II, and 13-BMC and 16-BMD beamlines at APS. We acknowledge DESY (Hamburg, Germany), a member of the Helmholtz

Association HGF, for the provision of experimental facilities. Our study was also performed as a part of the Early Science Program of the Centre for Molecular Water Science (CMWS) that is currently being set up at DESY. Use of 13-BMC is supported by NSF EAR-1634415 and EAR-1661511. Portions of this work were performed at HPCAT (Sector 16), Advanced Photon Source (APS), Argonne National Laboratory. HPCAT operations are supported by DOE-NNSA's Office of Experimental Sciences. The Advanced Photon Source is a U.S. Department of Energy (DOE) Office of Science User Facility operated for the DOE Office of Science by Argonne National Laboratory under Contract No. DE-AC02-06CH11357.

## Author contributions

Y.B. performed synchrotron experiments and data analysis with the help of H.H., H.-P.L., T.-Y.J., T.J.S., D.Z., and D.P. D.Y.K. and Y.H. contributed computational calculations. Y.L. supervised the research and worked on the manuscript with all authors.

## Competing interests

The authors declare no competing interests.
