## [Peer Review File · Nature Communications]

REVIEWER COMMENTS

Reviewer #1 (Remarks to the Author):

Remark to the Author

This paper presents the phase transition of pyrophyllite, one of the representative hydrous clay minerals in sedimentary rocks, under hydrous subduction environment in the alumina-silica-water system. In-situ synchrotron X-ray diffraction measurements combined with RH-DAC and theoretical approach using first-principles method showed that sequential breakdowns of pyrophyllite involving the formation of further hydrous minerals through the cold subduction of sediments. Based on the results, H₂O flux along subducting slab down to the lower mantle was discussed.

The discussion potentially provides some contributions to the debate regarding the mineralogy under hydrous conditions. However, from the perspective of water transport to the deep mantle, I do not think that this discussion has advanced much from previous works showing some hydrated minerals and their transitions under pressure (e.g., Hwang et al., 2021, Hwang et al., 2017). Furthermore, I am afraid that the experimental results obtained under water-saturated and very cold conditions may not be applicable to the natural system. Thus, I do not consider that it meets the requirement of covering a significant advance to be published in Nature communications.

Main comments are in the following.

(1) Geotherms along the subducting slab shown in Fig.1 are from South Mariana Trench and Kermadec Trench (W1300 model by Syracuse et al.). Since these slabs are extremely cold compared to the other regions, the data along these geotherms may not be applied to the discussion of the global water subduction. Furthermore, the experimental temperatures are lower than the slab surface through the depth (Fig.1). To convince the discussion in this study, further explanation especially for the relation between the temperature applied in this study and the geotherm of subducting slab in several regions, are needed.

(2) Because temperature conditions applied in the experiments were low to assert phase equilibrium for minerals, it is unclear that the reaction paths and structural evolution of pyrophyllite reflect actual phase transitions through subduction. It is reasonable to consider that phase change of pyrophyllite at temperatures below 200°C up to 5 GPa was kinetically inhibited. Actually, it seems that breakdown of pyrophyllite to gibbsite and diaspore occurs during temperature increase from 200°C to 400°C, at a constant pressure of 4 GPa. In such low temperatures, even stishovite also can contain large amount of hydrogen under metastable state (Spektor et al. 2011 PNAS). Discussion on the reaction kinetics is required to determine the depth of phase change through subduction in the geological time scale.

(3) I think H₂O-saturated condition applied in this experiment may be far from the actual Earth's interior. Why is free-H₂O component maintained at the slab surface in global? If we assume that free-H₂O is maintained in the slab for some reasons, water transportation is possible without hydrous mineral. Some discussion should be required to convince the water subduction proposed in this study.

(4) Line 78: "dry condition" is ambiguous expression since the referred previous paper studied the minerals with ASH system.

(5) Table 1 and S1: As written in line 615, temperature was monitored using thermocouples with maximum uncertainties of 3 K. Please explain how the error values shown in Table 1 are estimated.

(6) Fig.2(a): What do the gray and white symbols mean in this figure?

(7) Supplementary Table S3: A theoretical study (Mookherjee et al., 2016) was referred.

Reviewer #2 (Remarks to the Author):

The manuscript reports the reaction of pyrophyllite with water or NaCl-MgCl₂ solution at the P-T conditions of the subducting slabs. The study found that pyrophyllite breaks down into gibbsite+diaspore+coesite at ~135 km, followed by dehydration and water-preserving processes up to mantle transition zone (MTZ) and lower mantle. It is concluded that pyrophyllite could transport significant amount of H₂O along subduction zone to MTZ and the lower mantle. The experiments employing externally-heated diamond anvil cell appear to be properly performed and the data adequately analyzed. The results have important implications on the transport of H₂O into the Earth's lower mantle through the subducting slabs. I recommend publication of the manuscript after addressing the comments listed below.

Main comment:

- Pyrophyllite was loaded together with water or NaCl-MgCl₂ solution in the resistively-heated diamond anvil cell. So it was under water-saturated conditions. It is unclear whether sea water can reach to the depth below 100 km along the subducting slabs. Can authors elaborate the processes that could bring sea water to that depth so that it can react with pyrophyllite in the subducting slabs.

Minor comments:

- Line 82: change "resistive-heated diamond-anvil cell" to "resistively-heated diamond-anvil cell".

- Fig. 1: In Fig.1b, the second pattern from the top is for 5.5(3) GPa and 570 °C. Line 291 refers to Supplementary Figures 1-7 for detailed fitting results. However, the supplementary Figure 2 is for 4.5(5) GPa and 450 °C. Please replace the figure with the one for 5.5 GPa and 570 °C for consistency. Some major XRD peaks collected at 5.5 GPa and 570 °C in Fig. 1b are not labeled. For example, the peak at Q <1.0.

- Line 114 & supplementary Figs. 6-7: Stishovite was observed to form at below 255 km depth. Does stishovite formed during the reactions contain Al or H? This can be verified by comparing the unit cell parameters/volumes of the stishovite phases with those of pure SiO₂ stishovite in the literature. If stishovite contains water or H, it may be a major host of H₂O and thus influence the estimates of H₂O transport into the MTZ and lower mantle.

- Line 611: Replace "internal resistive electrical heaters" with "external resistive electrical heaters", as "internal heaters" are typically referred to as a mini-heater placed between diamond anvils in the sample chamber.

Response to Reviewers

Dear Reviewers,

We appreciate all of the insightful comments and constructive suggestions for our manuscript to Nature Communications (NCOMMS-23-10747) entitled “A role for subducting clays in the water transportation into the Earth’s lower mantle”. Please find attached our revised version of the paper (all the changes made in the revised manuscript **are marked in red**). Our point-by-point responses to all the comments and criticisms are summarized below (**the reviewers’ comments are marked in blue**).

Reviewer #1 (Remarks to the Author):

This paper presents the phase transition of pyrophyllite, one of the representative hydrous clay minerals in sedimentary rocks, under hydrous subduction environment in the alumina-silica-water system. In-situ synchrotron X-ray diffraction measurements combined with RH-DAC and theoretical approach using first-principles method showed that sequential breakdowns of pyrophyllite involving the formation of further hydrous minerals through the cold subduction of sediments. Based on the results, H₂O flux along subducting slab down to the lower mantle was discussed.

The discussion potentially provides some contributions to the debate regarding the mineralogy under hydrous conditions. However, from the perspective of water transport to the deep mantle, I do not think that this discussion has advanced much from previous works showing some hydrated minerals and their transitions under pressure (e.g., Hwang et al., 2021, Hwang et al., 2017). Furthermore, I afraid that the experimental results obtained under water-saturated and very cold conditions may not be applicable to the natural system. Thus, I do not consider that it meets the requirement of covering a significant advance to be published in Nature communications.

Reply: We appreciate the reviewer for the overall positive appraisal on the quality of our work. For this revision, we have performed additional in-situ HP-HT experiments to follow ‘moderately’ cold slab surface geotherms, e.g., Izu-Bonin subduction model, for our results to be applicable to a general cold subduction system. We have confirmed that pyrophyllite decomposes into diaspore + coesite assemblage as we have reported in the original manuscript which mainly followed the South Mariana and Kermadec subduction models. We advocate that our results contribute significant advancement to the previous findings as we have re-evaluated the overall role of clays, major components in subducting sedimentary layer, on the water transport capacity/efficiency and linked them down to the lower mantle, that is found to be comparable to the water transport by subducting crust.

Main comments are in the following.

(1) Geotherms along the subducting slab shown in Fig.1 are from South Mariana Trench and Kermadec Trench (W1300 model by Syracuse et al.). Since these slabs are extremely cold compared to the other regions, the data along these geotherms may not be applied to the discussion of the global water subduction. Furthermore, the experimental temperatures are lower than the slab surface through the depth (Fig.1). To convince the discussion in this study, further explanation especially for the relation between the temperature applied in this study and the geotherm of subducting slab in several regions, are needed.

Reply: As stated above, we have conducted additional HP-HT experiment up to 4.3(3) GPa and 600±50 °C to follow ‘moderately’ cold slab surface geotherms e.g., Izu-Bonin subduction model with ~6 °C/km. The result shows that pyrophyllite decomposes into diaspore + coesite assemblage, which is in line with the result from the original manuscript that simulated South Mariana and Kermadec subduction surface with < 5 °C/km. The new experimental details and results have been added in p.5, line 110-114 and p.6, line 136-142 together with revised Figure 1a, Supplementary Table 1, Supplementary Figs. 8 and 11, references #35-36.

(2) Because temperature conditions applied in the experiments were low to assert phase equilibrium for minerals, it is unclear that the reaction paths and structural evolution of pyrophyllite reflect actual phase transitions through subduction. It is reasonable to consider that phase change of pyrophyllite at temperatures below 200°C up to 5 GPa was kinetically inhibited. Actually, it seems that breakdown of pyrophyllite to gibbsite and diaspore occurs during temperature increase from 200°C to 400°C, at a constant pressure of 4 GPa. In such low temperatures, even stishovite also can contain large amount of hydrogen under metastable state (Spektor et al. 2011 PNAS). Discussion on the reaction kinetics is required to determine the depth of phase change through subduction in the geological time scale.

Reply: We believe the issue on the stability of pyrophyllite has been addressed, at least in part, by our additional experiment. We point out that the breakdown of pyrophyllite to gibbsite and diaspore occurs during temperature increase from 400 °C to 600 °C near 4 GPa (p.5-6, line 124-127 and line 136-139, revised Fig. 1). During our RH-DAC experiments, the samples were heated for about 12 hours (or more) to reach target temperature, where the sample temperature was kept for at least ~20–30 minutes to ensure no further changes are occurring. The RH-DAC setup that we used in this study offers the advantage of providing homogeneous and stable temperature across the entire sample chamber (Method section p.30-31, line 744-745 and 757-759, reference #96). We understand that the actual depths for phase transition could change when reaction kinetics are considered, together with other chemical/phase components existing in a real subducting layer. Although we have considered the composition of subduction fluid as an additional variable, i.e., using three different fluid compositions of pure water, salt water, and anhydrous silicone oil, to access their role as a catalyst as well as a reactant to affect kinetic barriers and breakdown scheme, further research is desired to fully understand the mineral/rock stabilities in a more realistic subduction environments (p.6 line 140-142, p.7, line 156-167, reference #42-47).

(3) I think H₂O-saturated condition applied in this experiment may be far from the actual Earth's interior. Why is free-H₂O component maintained at the slab surface in global? If we assume that free-H₂O is maintained in the slab for some reasons, water transportation is possible without hydrous mineral. Some discussion should be required to convincing the water subduction proposed in this study.

Reply: To address the issue on free water in a subduction system, we have added new sentences with references to showcase major fluid migration pathways and fluid-mineral interaction processes in a subduction zone. In the vicinity of subduction trench, water/seawater can be transported as sedimentary pore fluids (fluid-rich sediment) whereas in a deeper region down to > 100 km (beneath volcanic arcs and into the deeper mantle), fluid flux is channelized along open fractures and ductile shear zones. In line of this, we have revised the inset in Fig. 3a to illustrate the above two fluid migration pathways along relatively shallow and intermediate subduction interfaces (revised Fig. 3a, p.2-3, line 46-70, reference #5-12).

(4) Line 78: “dry condition” is ambiguous expression since the referred previous paper studied the minerals with ASH system.

Reply: We have deleted the phrase “under dry condition” to clarify the experimental conditions (p.5, line 104).

(5) Table 1 and S1: As written in line 615, temperature was monitored using thermocouples with maximum uncertainties of 3 K. Please explain how the error values shown in Table 1 are estimated.

Reply: We have added new sentences (with a new reference) to explain the estimation scheme of errors in temperature measurements in Table 1 and Supplementary Table 1. The overall uncertainty in the sample temperature has been estimated by the differences between the recorded thermocouple temperature and the calculated temperature based on the unit-cell volume and the known thermal expansion coefficient of standard materials (Method section p.30-31, line 749-752, reference #96).

(6) Fig.2(a): What do the gray and white symbols mean in this figure?

Reply: We have added the sentences “The reaction products from pyrophyllite and kaolinite are denoted by colored and white circles, respectively, while the other aluminosilicate minerals within the ASH system are marked with gray symbols.” (revised caption of Fig. 2).

(7) Supplementary Table S3: A theoretical study (Mookherjee et al., 2016) was referred.

Reply: We have replaced the theoretical study (Mookherjee et al., 2016) with more relevant experimental study (Komatsu et al., 2003) (revised Supplementary Table 3).

Reviewer #2 (Remarks to the Author):

The manuscript reports the reaction of pyrophyllite with water or NaCl-MgCl₂ solution at the P-T conditions of the subducting slabs. The study found that pyrophyllite breaks down into gibbsite+diaspore+coesite at ~135 km, followed by dehydration and water-preserving processes up to mantle transition zone (MTZ) and lower mantle. It is concluded that pyrophyllite could transport significant amount of H₂O along subduction zone to MTZ and the lower mantle. The experiments employing externally-heated diamond anvil cell appear to be properly performed and the data adequately analyzed. The results have important implications on the transport of H₂O into the Earth's lower mantle through the subducting slabs. I recommend publication of the manuscript after addressing the comments listed below.

Reply: We appreciate the overall positive appraisal of our work by the reviewer #2.

Main comment:

- Pyrophyllite was loaded together with water or NaCl-MgCl₂ solution in the resistively-heated diamond anvil cell. So it was under water-saturated conditions. It is unclear whether sea water can reach to the depth below 100 km along the subducting slabs. Can authors elaborate the processes that could bring sea water to that depth so that it can react with pyrophyllite in the subducting slabs.

Reply: We have added new sentences (with new references) to showcase major fluid migration pathways and fluid-mineral interaction processes in a subduction zone. In the vicinity of subduction trench, seawater can be transported as sedimentary pore fluids (fluid-rich sediment) whereas in a deeper region down to ~100 km (beneath volcanic arcs and into the deeper mantle), fluid flux is channelized along open fractures and ductile shear zones. In line of this, we have revised the inset in Fig. 3a to illustrate the above two fluid migration pathways along relatively shallow and intermediate subduction interfaces (revised Fig. 3a, p.2-3, line 46-70, reference #5-12). In fact, numerous studies demonstrated that major and trace elements (or solutes) dissolved in subduction zone fluids (SZFs) are recycled in the deep mantle. As NaCl and MgCl₂ are identified as important components in SZFs (Frezzotti & Ferrando, (2015)) as well as in seawater, we have considered the composition of subduction fluid as an additional variable, i.e., using three different fluid compositions of pure water, salt water, and anhydrous silicone oil to access their role as a catalyst as well as a reactant to affect kinetic barriers and breakdown scheme. Previous studies have further appraised the role of subduction fluid (deep recycling of seawater-derived fluids), as Holland and Ballentine (2006) revealed that subduction of sediment and seawater-dominated pore fluids play an important role in understanding the elemental abundance pattern of heavy noble gases (Ar, Kr, and Xe) in the mantle to be remarkably close to that of seawater (i.e., recycling of seawater is responsible for the observed heavy noble gas pattern in the convecting mantle as subduction occurs in a seawater-dominated fluid environment) (p.7, line 156-167, reference #42-47).

Minor comments:

- Line 82: change “resistive-heated diamond-anvil cell” to “resistively-heated diamond-anvil cell”.

Reply: We changed the phrase to “resistively-heated diamond anvil cell” (p.5, line 107).

- Fig. 1: In Fig.1b, the second pattern from the top is for 5.5(3) GPa and 570 °C. Line 291 refers to Supplementary Figures 1-7 for detailed fitting results. However, the supplementary Figure 2 is for 4.5(5) GPa and 450 °C. Please replace the figure with the one for 5.5 GPa and 570 °C for consistency. Some major XRD peaks collected at 5.5 GPa and 570 °C in Fig. 1b are not labeled. For example, the peak at $Q < 1.0$.

Reply: As pointed, we have replaced Supplementary Figure 2 with the correct one measured at 5.5 GPa and 570 °C and added the label in the XRD pattern in Figure 1b (new Supplementary Fig. 2 and revised Fig.1b).

- Line 114 & supplementary Figs. 6-7: Stishovite was observed to form at below 255 km depth. Does stishovite formed during the reactions contain Al or H? This can be verified by comparing the unit cell parameters/volumes of the stishovite phases with those of pure SiO₂ stishovite in the literature. If stishovite contains water or H, it may be a major host of H₂O and thus influence the estimates of H₂O transport into the MTZ and lower mantle.

Reply: We appreciate such an insightful comment by the reviewer #2. Accordingly, we have added new sentences (with new references) to address the issue on possible hydration in stishovite. Stishovite formed in our experiments shows differences in the refined unit cell volume by about +1–2 % compared to anhydrous SiO₂ at the corresponding HP-HT conditions (Angel et al., 2005; Kulik et al., 2018; Nishihara et al., 2005) (see Table below). However, recent experimental studies have established that hydrous stishovite can be formed at ~9–10 GPa and 350–550 °C while above 600 °C anhydrous stishovite is preferred (Spektor et al., 2011 and 2016; Nisr et al., 2017, references #39-41). The temperature range where stishovite formed in our experiments exceeds 550 °C, and the refined unit cell volume of the SiO₂ phase below 550 °C showed similar degree of expansion, which made us conclude that the apparent increase in the unit cell volume is due to systematic deviation within experimental uncertainty and the stishovite formed in our experiment is anhydrous (p.6-7, line 150-151).

Table | Comparison of the refined unit-cell volume of the SiO₂ phases to the values in the literature.

P-T	V (Å ³) ^a	V_{HPHT} (Å ³) ^{b,c}	$\Delta V/V_{HPHT}$ (%) ^d
5.5(3) GPa and 570±35 °C	529.64	521.64 ^b	1.53
6.9(3) GPa and 700±45 °C	528.61	516.16 ^b	2.41
7.6(4) GPa and 850±50 °C	528.15	513.86 ^b	2.78
6.3(4) GPa and 900±55 °C	531.54	519.50 ^b	2.32
13.6(10) GPa, 725±75 °C	45.98	45.25 ^c	1.62
20.7(10) GPa, 790±110 °C	45.30	44.33 ^c	2.20

^aRefined unit cell volume of the SiO₂ phase at P-T conditions in our experiments as in Supplementary Fig. 2–7

^bCalculated unit cell volume of coesite at the corresponding P-T conditions using P - V and V - T equation of state (Angel et al., 2005; Kulik et al., 2018)

(3rd-order Birch-Murnaghan EoS: $V_0 = 546.7(12)$ Å³, $K_{T0} = 97.4(6)$, $K'_{T0} = 4.3(2)$)

(Mie-Grüneisen-Debye EoS based on Suzuki model: $\theta_0 = 1004(43)$ K, $\gamma_0 = 0.271(16)$)

^cCalculated unit cell volume of stishovite at the corresponding P-T conditions using P - V - T equation of state (Nishihara et al., 2005)

(Mie-Grüneisen-Debye EoS: $V_0 = 46.547(12)$ Å³, $K_{T0} = 299(4)$, $K'_{T0} = 4$, $\theta_0 = 1160(120)$ K, $\gamma_0 = 1.33(6)$, $q = 6.1(8)$)

$$^d\Delta V/V_{HRHT} \% = (V - V_{HRHT}) / V_{HRHT} \times 100$$

- Line 611: Replace "internal resistive electrical heaters" with "external resistive electrical heaters", as "internal heaters" are typically referred to as a mini-heater placed between diamond anvils in the sample chamber.

Reply: As suggested, we have replaced the phrase (p.30, line 742).

Based on the above, we sincerely believe that we have addressed all the criticisms and comments of the reviewers and therefore request our revised manuscript to be considered for publication in Nature Communications.

REVIEWERS' COMMENTS

Reviewer #2 (Remarks to the Author):

The revised manuscript has addressed my major comment on the fluid migration pathways in subduction zones. The authors have also adequately made relevant revisions in response to the minor comments from the first round of review. Additional experiments were performed at higher temperature up to 600 °C to address the main comments on the discrepancy between the highest temperature achieved in experiments and the real slab surface temperature at depth. Therefore, I do not have any further comments on the revised manuscript and recommend its publication in Nature Communications.

Reviewer #3 (Remarks to the Author):

This paper describes the phase transformation sequences of the ASH system within cold subducting slabs starting from pyrophyllite, and their implication for deep water transport within the Earth. I believe the paper is worth to be considered for publication in Nature Communication. Here are my comments on the author's reply to Reviewer #1 comments.

(1) Author's Reply: As stated above, we have conducted additional HP-HT experiment up to 4.3(3) GPa and 600±50 °C to follow 'moderately' cold slab surface geotherms e.g., Izu-Bonin subduction model with ~6 °C/km. The result shows that pyrophyllite decomposes into diaspore + coesite assemblage, which is in line with the result from the original manuscript that simulated South Mariana and Kermadec subduction surface with < 5 °C/km. The new experimental details and results have been added in p.5, line 110-114 and p.6, line 136-142 together with revised Figure 1a, Supplementary Table 1, Supplementary Figs. 8 and 11, references #35-36.

(1) Reviewer comment: I consider that the additional HP-HT experiment effectively reinforced the result and the discussion in the original manuscript. To clarify that the "moderately cold slab surface geotherms, i.e., following the thermal model of Izu-Bonin subduction" (lines 136 to 137) were independently analyzed from that of the coldest slab ones, it is necessary for the revised manuscript to explicitly note that the experimental P-T path for "up to 4.3(3) GPa and 600±50 °C" was independently analyzed from that for "~23 GPa and 700-900 °C following the cold subduction geotherms of the South Mariana and Kermadec thermal Models" (lines 122 to 124).

(2) Author's Reply: We believe the issue on the stability of pyrophyllite has been addressed, at least in part, by our additional experiment. We point out that the breakdown of pyrophyllite to gibbsite and diaspore occurs during temperature increase from 400 °C to 600 °C near 4 GPa (p.5-6, line 124-127 and line 136-139, revised Fig. 1). During our RH-DAC experiments, the samples were heated for about 12 hours (or more) to reach target temperature, where the sample temperature was kept for at least ~20–30 minutes to ensure no further changes are occurring. The RH-DAC setup that we used in this study offers the advantage of providing homogeneous and stable temperature across the entire sample chamber (Method section p.30-31, line 744-745 and 757- 759, reference #96). We understand that the

actual depths for phase transition could change when reaction kinetics are considered, together with other chemical/phase components existing in a real subducting layer. Although we have considered the composition of subduction fluid as an additional variable, i.e., using three different fluid compositions of pure water, salt water, and anhydrous silicone oil, to assess their role as a catalyst as well as a reactant to affect kinetic barriers and breakdown scheme, further research is desired to fully understand the mineral/rock stabilities in a more realistic subduction environments (p.6 line 140-142, p.7, line 156-167, reference #42-47).

(2) Reviewer comment: The response seems satisfactory to address the given issue, and the manuscript was appropriately revised.

(3) Author's Reply: To address the issue on free water in a subduction system, we have added new sentences with references to showcase major fluid migration pathways and fluid-mineral interaction processes in a subduction zone. In the vicinity of subduction trench, water/seawater can be transported as sedimentary pore fluids (fluid-rich sediment) whereas in a deeper region down to > 100 km (beneath volcanic arcs and into the deeper mantle), fluid flux is channelized along open fractures and ductile shear zones. In line of this, we have revised the inset in Fig. 3a to illustrate the above two fluid migration pathways along relatively shallow and intermediate subduction interfaces (revised Fig. 3a, p.2-3, line 46-70, reference #5-12).

(3) Reviewer comment: The response seems satisfactory to address the given issue, and the manuscript and Figure 3 were appropriately revised.

The other remarks were answered as just requested by the referee #1.

Additional reviewer comment: "SFZs" should rather be "SZFs" (if following Mannning and Frezzotti, 2020).

Response to Reviewers

Dear Reviewer,

We appreciate all of the valuable comments and constructive suggestions for our manuscript to Nature Communications (NCOMMS-23-10747A) entitled “A role for subducting clays in the water transportation into the Earth’s lower mantle”. Please find attached the 2nd revised version of the paper (all the changes made in the revised manuscript **are marked in red**). Our point-by-point responses to all the comments are summarized below (the reviewers’ comments **are marked in blue**).

Reviewer #2 (Remarks to the Author):

The revised manuscript has addressed my major comment on the fluid migration pathways in subduction zones. The authors have also adequately made relevant revisions in response to the minor comments from the first round of review. Additional experiments were performed at higher temperature up to 600 °C to address the main comments on the discrepancy between the highest temperature achieved in experiments and the real slab surface temperature at depth. Therefore, I do not have any further comments on the revised manuscript and recommend its publication in Nature Communications.

Reply: We thank Reviewer #2 for the appreciation of our work and efforts to improve our manuscript. We are very pleased to hear from Reviewer #2 that our revised manuscript has addressed the major concerns regarding the fluid migration pathways in subduction zones, as well as the minor comments raised in the first round review. We appreciate very much the encouraging comments and support of Reviewer #2 for publication of our manuscript in Nature Communications.

Reviewer #3 (Remarks to the Author):

This paper describes the phase transformation sequences of the ASH system within cold subducting slabs starting from pyrophyllite, and their implication for deep water transport within the Earth. I believe the paper is worth to be considered for publication in Nature Communication. Here are my comments on the author's reply to Reviewer #1 comments.

Reply: We appreciate the encouraging remarks and insightful comments by Reviewer #3. We are thankful for the review and recommendation for publication of our manuscript in Nature Communications. We hope we have addressed all the concerns raised by reviewer #3 in this revision. Our point-by-point responses to the comments are as follows.

(1) Author's Reply: As stated above, we have conducted additional HP-HT experiment up to 4.3(3) GPa and 600±50 °C to follow ‘moderately’ cold slab surface geotherms e.g., Izu-Bonin

subduction model with ~ 6 °C/km. The result shows that pyrophyllite decomposes into diaspore + coesite assemblage, which is in line with the result from the original manuscript that simulated South Mariana and Kermadec subduction surface with < 5 °C/km. The new experimental details and results have been added in p.5, line 110-114 and p.6, line 136-142 together with revised Figure 1a, Supplementary Table 1, Supplementary Figs. 8 and 11, references #35-36.

(1) Reviewer comment: I consider that the additional HP-HT experiment effectively reinforced the result and the discussion in the original manuscript. To clarify that the "moderately cold slab surface geotherms, i.e., following the thermal model of Izu-Bonin subduction" (lines 136 to 137) were independently analyzed from that of the coldest slab ones, it is necessary for the revised manuscript to explicitly note that the experimental P-T path for "up to 4.3(3) GPa and 600 ± 50 °C" was independently analyzed from that for " ~ 23 GPa and 700-900 °C following the cold subduction geotherms of the South Mariana and Kermadec thermal Models" (lines 122 to 124).

Reply: We have revised the manuscript to improve the clarity regarding the additional experiment under "moderately cold slab surface geotherms (Izu-Bonin thermal models)" by adding "Izu-Bonin" and "for which data have been independently measured and analyzed" in p. 6 line 130-131 and line 144-145, respectively.

(2) Author's Reply: We believe the issue on the stability of pyrophyllite has been addressed, at least in part, by our additional experiment. We point out that the breakdown of pyrophyllite to gibbsite and diaspore occurs during temperature increase from 400 °C to 600 °C near 4 GPa (p.5-6, line 124-127 and line 136-139, revised Fig. 1). During our RH-DAC experiments, the samples were heated for about 12 hours (or more) to reach target temperature, where the sample temperature was kept for at least ~ 20 –30 minutes to ensure no further changes are occurring. The RH-DAC setup that we used in this study offers the advantage of providing homogeneous and stable temperature across the entire sample chamber (Method section p.30-31, line 744-745 and 757- 759, reference #96). We understand that the actual depths for phase transition could change when reaction kinetics are considered, together with other chemical/phase components existing in a real subducting layer. Although we have considered the composition of subduction fluid as an additional variable, i.e., using three different fluid compositions of pure water, salt water, and anhydrous silicone oil, to access their role as a catalyst as well as a reactant to affect kinetic barriers and breakdown scheme, further research is desired to fully understand the mineral/rock stabilities in a more realistic subduction environments (p.6 line 140-142, p.7, line 156-167, reference #42-47).

(2) Reviewer comment: The response seems satisfactory to address the given issue, and the manuscript was appropriately revised.

Reply: We are grateful for acknowledging that our response and the revised manuscript have

addressed the given issue.

(3) Author's Reply: To address the issue on free water in a subduction system, we have added new sentences with references to showcase major fluid migration pathways and fluid-mineral interaction processes in a subduction zone. In the vicinity of subduction trench, water/seawater can be transported as sedimentary pore fluids (fluid-rich sediment) whereas in a deeper region down to > 100 km (beneath volcanic arcs and into the deeper mantle), fluid flux is channelized along open fractures and ductile shear zones. In line of this, we have revised the inset in Fig. 3a to illustrate the above two fluid migration pathways along relatively shallow and intermediate subduction interfaces (revised Fig. 3a, p.2-3, line 46-70, reference #5-12).

(3) Reviewer comment: The response seems satisfactory to address the given issue, and the manuscript and Figure 3 were appropriately revised.

The other remarks were answered as just requested by the referee #1.

Reply: We appreciate the comments of Reviewer #3 on our efforts to address the issues in the revised manuscript and Figure 3 (currently Figure 4).

Additional reviewer comment: "SFZs" should rather be "SZFs" (if following Manning and Frezzotti, 2020).

Reply: We have corrected "SFZs" to "SZFs" in the manuscript (p. 7 line 167-168).

Based on the above, we sincerely request our manuscript to be published in Nature Communications.